Using an integrative taxonomic approach to delimit a sibling species, Mycetomoellerius mikromelanos sp. nov. (Formicidae: Attini: Attina)

Cardenas Cody Raul cardenas.61@buckeyemail.osu.edu 1
Luo Amy Rongyan 2
Jones Tappey H. 3
Schultz Ted R. 4
Adams Rachelle M.M. adams.1970@osu.edu 1 4
1 Department of Evolution, Ecology and Organismal Biology, The Ohio State University , Columbus , OH , United States of America
2 Department of Ecology & Evolutionary Biology, University of Tennessee Knoxville , Knoxville , TN , United States of America
3 Department of Chemistry, Virginia Military Institute , Lexington , VA , United States of America
4 Department of Entomology, National Museum of Natural History, Smithsonian Institution , Washington , District of Colombia , United States of America
Gillespie Joseph
Electronic publication date: 2021 Jun 24
Publication date: 2021
Volume: 9
Electronic Location ID: e11622
Received 2020 Dec 16; Accepted 2021 May 26
Copyright: ©2021 Cardenas et al.
Copyright year: 2021
Copyright holder: Cardenas et al.
License: This is an open access article distributed under the terms of the Creative Commons Attribution License, which permits unrestricted use, distribution, reproduction and adaptation in any medium and for any purpose provided that it is properly attributed. For attribution, the original author(s), title, publication source (PeerJ) and either DOI or URL of the article must be cited.
License URL: https://creativecommons.org/licenses/by/4.0/

Keywords: Integrative taxonomy, Cryptic species, Formicidae, Myrmicinae, Attine, Attina, Fungus-growing ants

Funding: National Science Foundation DEB 1927224 1654829 Ted Schultz was supported by the National Science Foundation grants DEB 1927224 and 1654829. The funders had no role in study design, data collection and analysis, decision to publish, or preparation of the manuscript.

==============================
The fungus-growing ant Mycetomoellerius (previously Trachymyrmex) zeteki (Weber 1940) has been the focus of a wide range of studies examining symbiotic partners, garden pathogens, mating frequencies, and genomics. This is in part due to the ease of collecting colonies from creek embankments and its high abundance in the Panama Canal region. The original description was based on samples collected on Barro Colorado Island (BCI), Panama. However, most subsequent studies have sampled populations on the mainland 15 km southeast of BCI. Herein we show that two sibling ant species live in sympatry on the mainland: Mycetomoellerius mikromelanos Cardenas, Schultz, & Adams and M. zeteki. This distinction was originally based on behavioral differences of workers in the field and on queen morphology (M. mikromelanos workers and queens are smaller and black while those of M. zeteki are larger and red). Authors frequently refer to either species as “M. cf. zeteki,” indicating uncertainty about identity. We used an integrative taxonomic approach to resolve this, examining worker behavior, chemical profiles of worker volatiles, molecular markers, and morphology of all castes. For the latter, we used conventional taxonomic indicators from nine measurements, six extrapolated indices, and morphological characters. We document a new observation of a Diapriinae (Hymenoptera: Diapriidae) parasitoid wasp parasitizing M. zeteki. Finally, we discuss the importance of vouchering in dependable, accessible museum collections and provide a table of previously published papers to clarify the usage of the name T. zeteki. We found that most reports of M. zeteki or M. cf. zeteki—including a genome—actually refer to the new species M. mikromelanos.

Introduction

Fungus-growing ants (Hymenoptera: Formicidae: Attini: Attina; Ward et al., 2015), referred to as “attine” ants, cultivate mutualistic fungus gardens using sophisticated agricultural practices (Weber, 1958a). This clade of 240 extant described species has been tending and feeding cultivated fungi for ca. 60 million years (Branstetter et al., 2017). Because fungus-growing ants have been focal taxa of studies in evolutionary biology, including mating systems (Baer & Boomsma, 2004; Boomsma, 2007), symbiotic networks (Mueller, Rehner & Schultz, 1998; Currie, Mueller & Malloch, 1999), social parasitism (Adams et al., 2013), host fidelity (Mehdiabadi et al., 2012), and genome evolution (Nygaard et al., 2016), it is imperative that the taxonomy of attine ants accurately reflects their evolutionary history. Diverse studies indicate the existence of many undescribed species (Schultz & Meier, 1995; Schultz, Bekkevold & Boomsma, 1998; Rabeling et al., 2007; Schultz & Brady, 2008; Mehdiabadi et al., 2012; Ješovnik et al., 2013; Sosa-Calvo et al., 2018; Solomon et al., 2019) and alpha-taxonomic work has been steadily carried out by many taxonomists (Mayhé-Nunes & Brandão, 2002; Mayhé-Nunes & Brandão, 2005; Mayhé-Nunes & Brandão, 2007; Sosa-Calvo & Schultz, 2010; Ješovnik et al., 2013; Rabeling et al., 2015; Ješovnik & Schultz, 2017; Sosa-Calvo et al., 2017; Sosa-Calvo et al., 2018). In fact an average of 2.4 new attine species have been described per year from 1995 to 2019 (Table S1, e.g., Schultz et al., 2002; Ješovnik et al., 2013; Sánchez-Peña et al., 2017; Sosa-Calvo et al., 2018; Cristiano et al., 2020).

Taxonomists have informally split the attines into lower and higher fungus-growing ants based on varying systems of obligate fungus-farming agriculture (Schultz & Brady, 2008; Branstetter et al., 2017). The lower attines cultivate a diversity of fungal cultivar lineages, while the higher attines generally cultivate more closely related lineages of fungal species including Leucoagaricus gongylophorus (Möller) Singer, 1986. The most derived and familiar higher attine genera consist of the leaf-cutting ants, Atta Fabricus, 1804, Acromyrmex Mayr, 1865, and Amoimyrmex Cristiano et al., 2020 which largely cut fresh plant material for their gardens. However, the other higher attine genera consist of Sericomyrmex Mayr, 1865, Trachymyrmex Forel, 1983, Xerolitor Sosa-Calvo et al., 2018, Mycetomoellerius Solomon et al., 2019, and Paratrachymyrmex Solomon et al., 2019. These non-leaf-cutting higher attines, referred to as higher attines hereafter, are phylogenetically intermediate between the lower-attine and leaf-cutting ants (Brandão & Mayhé-Nunes, 2007).

Higher attine ants share natural history traits with both the lower attines and leaf-cutting ants. Similar to leaf-cutting ants, some higher attines have also been observed cutting fresh plant material for their gardens (Weber, 1972; Schultz & Meier, 1995; Leal & Oliveira, 2000; Mayhé-Nunes & Brandão, 2005; Brandão & Mayhé-Nunes, 2007). Otherwise, much like lower attines, higher attines typically harvest fallen flowers, fruits, leaves, small twigs, seeds, and caterpillar frass (Lizidatti, 2006; De Fine Licht & Boomsma, 2010; Ronque, Feitosa & Oliveira, 2019). Unlike lower-attine workers that are typically monomorphic, workers in Mycetomoellerius, Paratrachymyrmex, and Trachymyrmex tend to be weakly polymorphic (Weber, 1958a; Beshers & Traniello, 1996; Brandão & Mayhé-Nunes, 2007; Rabeling et al., 2007). It is this variability in worker morphology, coupled with species descriptions based on a few workers (Weber, 1940), sampling bias (see Mueller et al., 2018), and inconsistent voucher deposition that have led to incorrect or incomplete species identifications (Appendix Table S1).

The genus Mycetomoellerius is represented by 31 recognized species, distributed throughout Central and South America. Taxonomic clarity for this and related genera is needed as there are likely multiple new species, including a sister clade to the well-studied species M. zeteki (Weber, 1940; junior synonym M. balboai; see also Solomon et al., 2019). Mycetomoellerius zeteki is abundant and easily collected in the Panama Canal Zone and has been included in a large breadth of work (Appendix Table S1). Notable research includes the discovery and function of actinomycete bacteria in the fungus-growing ants (Currie, Mueller & Malloch, 1999; Mueller et al., 2008), description of the evolutionary transition from single to multiple mating in the fungus-growing ants (Villesen et al., 2002), and the reciprocal evolution of ant and fungal genomes in the fungus-growing ant symbiosis (Nygaard et al., 2016). Despite this attention to its biology, M. zeteki has remained taxonomically ambiguous. For example, in a phylogenetic analysis of actinomycetes bacteria, three samples form a polytomy containing M. sp. ‘Funnel’, an undetermined Mycetomoellerius sp., and M. zeteki sensu stricto (Cafaro & Currie, 2005). It has been speculated that the current definition of M. zeteki may include cryptic species based on behavioral (Adams et al., 2012b), morphological (Adams et al., 2012b), molecular (Solomon et al., 2019), and chemical differences (Adams, Jones & Jeter, 2010; Adams et al., 2012a).

This uncertainty surrounding M. cf. zeteki has ramifications given its significant historical contributions to fungus-growing ant research (Appendix Table S1). To resolve this, we use an integrative approach to clarify the taxonomy of M. zeteki by reexamining morphological characters, comparing old and new collections, examining morphometrics, adapting a comparative behavioral method for worker tempo, and chemically analyzing worker volatile compounds. Based on these diverse data, we recognize two species: M. zeteki and Mycetomoellerius mikromelanos sp. nov. Cardenas, Schultz, & Adams. We provide a diagnosis and description of M. mikromelanos sp. nov., describe the M. zeteki gyne wings and the morphological characters of M. zeteki males, determine the identity of the published M. zeteki genome, suggest corrections for the misidentification of voucher specimens in published research, and discuss the implications of our improved species-level definitions.

Materials & Methods

Samples and collections

Colonies of M. mikromelanos sp. nov. (31 colonies) and M. zeteki (16 colonies) were collected at the start of the wet season in 2017 and 2018 in the Canal Zone of the Republic of Panama (9.12007, -79.7317). Colony collection and fieldwork was approved by The Smithsonian Tropical Research Institute as part of the “Behavioral Ecology and Systematics of the Fungus-growing Ants and Their Symbionts (#4056)” project and the Autoridad Nacional del Ambiente y el Mar (Permiso de Colecta Científica 2017: SPO-17-173, 2018: SE/AB-1-18). Samples were collected by excavating only the first (i.e., upper) chamber of the nest to ensure colony survival. Of those excavated in 2018, 16 of 30 colonies were collected into five-dram vials (BioQuip, Cat. 8905 California, United States) and transferred to Petri dishes lined with moist cotton fiber for observations while in Panama. Vouchers of ca. 10 or more workers and fungus gardens from each nest were collected in 95% EtOH. Live colonies were brought back to The Ohio State University to a United States Department of Agriculture Animal and Plant Health Inspection Service Approved Facility (OSU; Columbus Ohio, USA; APHIS permit P526P-16-02785; facility #4036), where they were transferred to permanent nest boxes (as in Sosa-Calvo et al., 2015). When colonies were excavated, many contained workers, and male and female reproductive’s. We refer to the winged female reproductive caste as gynes unless otherwise noted. Few actual queens were examined. In nearly all cases, if a gyne or queen was present, there are also workers from the same colony (except one specimen from Jack Longino).

Taxonomy & morphometrics

We used a Wild M-5 microscope equipped with an ocular micrometer to examine specimens for morphological characters that unambiguously separate the two species. We also took morphological measurements of 171 workers (n = 54 M. zeteki, n = 117 M. mikromelanos sp. nov.), 53 queens and gynes (n = 28 M. zeteki, n = 25 M. mikromelanos sp. nov.), and 43 males (n = 22 M. zeteki, n = 21 M. mikromelanos sp. nov.) using standard morphometrics (Table 1). We included two synonymized M. balboai syntypes (‘cotypes’) and one additional specimen identified as M. balboai. Including this junior synonym of M. zeteki (Weber, 1958b) was necessary to confirm that M. mikromelanos is not M. balboai. Upon confirmation, these samples were included as M. zeteki in further analyses. Terminology for the temple and malar areas follows that of Boudinot, Sumnicht & Adams (2013) and for sculpturing that of Harris (1979). Type and voucher specimens of material examined are deposited at the United States National Museum (USNM), Museum of Zoology of the University of São Paulo (MZSP), Smithsonian Tropical Research Institute (STRI), and The Ohio State University Museum of Biological Diversity Triplehorn Insect Collection (OSUC).

Table 1 Acronyms of standard measurements and indices used for morphology and morphometrics.

Eyes are included in HW for males.

Standard measurements	
HW	Head Width, maximum width of the head, in full-face view.	
HL	Head Length, maximum length diagonal from the anterior margin of the clypeus to the tip of the posterior margin of the head, in full-face view.	
SL	Scape Length, the maximum length of the scape from apex to basal flange, in dorsal view. Not including the basal condyle and neck.	
EL	Eye Length, maximum length of the eye, in lateral view.	
FL	Frontal Lobes, maximum length form the margins of the frontal lobes, in face view.	
ML	Mesosoma Length, diagonal distance from basal inflection of the anterior pronotal flange to the posterior most extension of the metapleural gland, in lateral view.	
PL	Petiole Length, length from the metapleural gland to the insertion of the post-petiole, in lateral view.	
PPL	Post-petiole Length, length from the anterior insertion point of the petiole to the anterior most point of the tergite-sternite gastral suture, in lateral view.	
GL	Gaster Length, length from the anterior most point of the tergite-sternite gastral suture to the furthest posterior point, in lateral view.	
Indices	
CI	Cephalic Index, 100 × HW/HL	
EI	Eye Index, 100 × EL/HL	
SI	Scape Index, 100 × SL/HW	
FLI	Frontal Lobe Index, 100 × FL/HW	
WaL	Waist Length, PL + PPL	
TL	Total Length, HL + ML + PL + PPL + GL	

The electronic version of this article in Portable Document Format (PDF) will represent a published work according to the International Commission on Zoological Nomenclature (ICZN), and hence the new names contained in the electronic version are effectively published under that Code from the electronic edition alone. This published work and the nomenclatural acts it contains have been registered in ZooBank, the online registration system for the ICZN. The ZooBank LSIDs (Life Science Identifiers) can be resolved and the associated information viewed through any standard web browser by appending the LSID to the prefix http://zoobank.org/. The LSID for this publication is: urn:lsid:zoobank.org:pub:737E04E5-5A8F-48F6-BE32-ADC1028927B6. The online version of this work is archived and available from the following digital repositories: PeerJ, PubMed Central and CLOCKSS.

We partitioned specimens by caste and tested the assumption of normality for each morphometric character with a Shapiro–Wilks test. We used a Welch’s t-test for normally distributed and a Wilcoxon Rank Sum test for non-normally distributed variables to test the null hypothesis of equal means and differences in range between both species. In the Wilcoxon Rank Sum Test there were ties in the data, so exact p-values could not be calculated for all castes. Both the normality testing and difference of means was performed in the base R package ‘stats’ (R CoreTeam, 2017). To reduce the risk of Type I error, only measurements with a Bonferroni corrected P-value (p < 0.003) were included.

With our retained variables, we performed non-metric multidimensional scaling (NMDS) with the vegan R package, using the ‘metaMDS’ function (Oksanen et al., 2019). This function calculates the Bray-Curtis distances, applies a square root transformation, and scales the distance measures down to k dimensions. We set k = 2 and searched for a solution with 1,000 random starts (see McCune & Grace, 2002; Glon et al., 2019; Oksanen et al., 2019). We subsequently produced a diagnostic Shepard plot with the ‘stressplot’ command from vegan. We considered our reduced dimensions acceptable if our transformed data reasonably fit the regression of the Shepard plot and if stress scores were <0.20 (McCune & Grace, 2002). We generated NMDS plots with characters plotted as vectors and 95% confidence ellipses for each species.

Behavioral assay

We adapted the novel environment assay (Chapman et al., 2011) to examine the tempo, i.e., activity level, of workers of M. zeteki and M. mikromelanos sp. nov. We subsampled four colonies of each species with five trials per colony. Single workers were selected from the foraging chamber and placed in the center of a nine cm Petri dish lined with one cm2 grid paper. The ant was immediately covered with one quarter of a 4.5 cm weigh boat (referred to as “refuge” hereafter). Five-minute trials were recorded with a Sony DCR-PC109 camera, digitized from the cassette tape, and scored using Solomon Coder (Péter, 2017). We measured (1) time to initially emerge from the refuge, (2) number of squares the ant entered, and (3) time spent under the refuge after the initial emergence. To analyze the change in tempo over the trial, we produced a ratio of squares entered to time spent entering squares (i.e., not under the refuge): New Squares/(300 s - Time to Exit Refuge - Time Under Refuge - Time on Refuge) = Tempo.

To test whether tempo differed between species we used a generalized linear mixed model (GLMM) with the ‘lme4’ package (Bates et al., 2015) in R (R CoreTeam, 2017). This experiment included multiple workers from the same colony, which are not independent from one another. To account for the blocked design (i.e., multiple data points from the same colony), we included the workers’ colony of origin as a random effect in the model. We compared a linear mixed model and multiple GLMMs using gaussian and gamma distribution families with the log, identity, and inverse link functions. To confirm the fit of the model, we first checked the normality of the residuals using a QQ-plot and Shapiro–Wilk test. We then checked for linearity and homoscedasticity by plotting the residuals and fitted values.

Phylogenetic analysis

We used sequence data published in Solomon et al. (2019); available on Dryad DOI: 10.5061/dryad.2p7r771) to confirm the identity of the published genome (Nygaard et al., 2016). We used sequences of M. zeteki, M. mikromelanos sp. nov. (listed as Mycetomoellerius n. sp. RMMA in Solomon et al., 2019; Table S2), and M. turrifex (Wheeler, 1903) from the dataset of Solomon et al. (2019) and aligned them in Geneious (version R9; Biomatters Limited, Auckland, New Zealand). We used BLAST with blastn and megablast (Altschul et al., 1990; Zhang et al., 2000; Morgulis et al., 2008) to identify quality gene regions in the published genome (Nygaard et al., 2016; GenBank accession: GCA_001594055.1). The gene for COI was removed from the analysis because COI data were missing for a subset of individuals in the data of Solomon et al. (2019). Megablast found no alignments and blastn found multiple scaffolds with high query cover (see Results and Table S3). In Geneious, we mapped our samples to the identified reference genome scaffolds and trimmed the areas of the scaffold that did not align. Once aligned, we concatenated our data into a multi-locus dataset with SequenceMatrix 1.8 (Vaidya, Lohman & Meier, 2011) for phylogenetic analysis. The four genes used are elongation factor 1-alpha F1 (EF1α -F1 1,074 bp), elongation factor 1-alpha F2 (EF1α-F2 434 bp), long-wavelength rhodopsin (LwRh 455 bp), and wingless (WG 702 bp).

For our phylogenetic analysis, we used ModelFinder (Kalyaanamoorthy et al., 2017) in IQ-TREE (version 1.6.10; Nguyen et al., 2015) to determine the best evolutionary model for each gene. The partitions with the most similar and likely models were merged in IQ-TREE and used to construct a maximum-likelihood phylogeny with M. turrifex as the outgroup and 10,000 ultrafast bootstraps (UFboot2; Hoang et al., 2018). Our resulting consensus tree was annotated in FigTree (version 1.4.3; Rambaut, 2016) and edited in Adobe Illustrator.

Chemical analysis

Volatile compounds were extracted from workers (as in Hamilton et al., 2018) sampled from lab-maintained colonies of M. mikromelanos sp. nov. (n = 6 colonies) and M. zeteki (n = 4 colonies). Samples of 4–10 individuals per colony were placed in HPLC grade methanol solvent. Whole ants from the same colony, or trisected ants (head, thorax, gaster), were placed in separate glass vials with 40–100 µL of solvent. Trisections were used to identify where the most abundant compounds were found and whole specimen extractions confirmed the presence of the compounds. Tools used for trisections were rinsed with ethanol, methanol, and pentane between trisection to prevent cross-contamination. Samples were stored at −20 °C until analysis by gas-chromatography mass-spectrometry (GC-MS). Reported compounds were found in at least trace amounts in two or more extracts of workers of the same species.

Samples of extracts were analyzed at the Virginia Military Institute with gas chromatography–mass spectrometry (GC–MS; as in Hamilton et al., 2018) using a Shimadzu QP-2010 GC–MS equipped with an RTX-5, 30 m × 0.25 mm i.d. column. The carrier gas was helium with a constant flow of 1 ml/min. The temperature program was from 60 to 250 °C changing 10 °C/min and held at the upper temperature for 20 min. The mass spectrometer was operated in EI mode at 70 eV, and scanning was set to 40 to 450 AMU at 1.5 scans/s. Peaks on chromatograms were identified by database search (NIST Mass Spectral Data base, V.2, US Department of Commerce, Gaithersburg, MD), published literature spectra, and by direct comparison with commercially available authentic samples. We standardized our resulting compounds for comparison. For each sample, ratios from the chromatogram peaks were converted to proportions and visualized in Adobe Illustrator.

Literature review

We conducted a literature review for all papers referencing M. zeteki or M. cf. zeteki to identify potentially misnamed species. Using the research databases Web of Knowledge (Clarivate Analytics, Massachusetts, United States), antweb.org (California Academy of Sciences, California, United States), hol.osu.edu (C.A. Triplehorn Insect Collection, Ohio, United States), and personal literature collections, we reviewed papers that were found by the search criterion “Trachymyrmex zeteki”, “Trachymyrmex cf. zeteki”, “T. zeteki”, “T. cf. zeteki”, “zeteki”, and “cf. zeteki”. We then selected articles that included M. cf. zeteki or M. zeteki as their focal research organism and recorded those that reported the deposition of voucher specimens. We disregarded research articles that did not use physical specimens (e.g., data from molecular databases).

Results

Morphometrics

Nearly all measurement means (Welch’s), and ranges (Wilcoxon) are different between the two species (Table 2). The samples of the junior synonym M. balboai are within the ranges of M. zeteki samples (see Table S4) and are morphologically similar to the M. zeteki type specimen. M. mikromelanos sp. nov. is on average smaller than M. zeteki except in the case of the frontal lobe index (FLI). Due to non-significant differences, FLI was excluded from analyses of males and gynes. We observed some overlap in the range of measurements for workers and for males between M. mikromelanos sp. nov. and M. zeteki. In contrast, gynes are very distinct with few overlapping ranges (Table 2).

WORKERS: For our worker partition, all 15 characters were significantly different between species (p < 0.003; Table 2). Our NMDS converged on a two-dimensional solution with an acceptable stress level (stress = 0.1288) and the Sheppard plot showed good association around the regression line (non-metric fit R2 = 0.983; linear fit R2 = 0.933; see Fig. S1a). The resulting NMDS plot shows some overlap between the ellipses, although each species forms a distinct cluster with few outliers (Fig. 1A). The vectors for head width (HW), scape index (SI), and petiole length (PL) showed the most strength and direction in the measurements relative to the NMDS axes (Fig. 1A). Additionally, the type specimens for M. mikromelanos sp. nov. and M. zeteki plotted within their own ellipses (Fig. 1A). While the M. mikromelanos sp. nov. type and paratype specimens fall within the overlap of ellipses for both species, they remain morphologically distinct (see diagnosis and description). For M. mikromelanos sp. nov., SI and FLI explain separation from the M. zeteki cluster; while HW, eye length (EL), and frontal lobe FL explain separation from M. mikromelanos sp. nov. for M. zeteki. However, PL and waist length (WaL) best explain variation within clusters along the Y axis. Lastly, the two synonymized M. balboai syntype (‘cotype’) samples fall well within the M. zeteki ellipses.

GYNES: For the gyne partition, all but FLI (p = 0.6110) were significantly different between species (p < 0.003; Table 2). The NMDS converged on a two-dimensional solution with a robust stress level (stress = 0.1119), and the Shepard plot showed a strong association around the regression line with a single outlier (non-metric fit R2 = 0.986; linear fit R2 = 0.941; see Fig. S1b). The NMDS plot showed M. mikromelanos sp. nov. and M. zeteki each forming distinct clusters with two outliers (Fig. 1B). The M. mikromelanos sp. nov. paratype gyne fell well within the M. mikromelanos sp. nov. cluster (Fig. 1B). The vectors EL, SI, and PL showed the most strength in directionality of the measurements relative to the NMDS axes (Fig. 1B).

Table 2 Select morphometrics of all castes.

Partitioned mean values, standard errors (SE), minimum and maximum standard measurements in millimeters, except indices (see Table 1) for workers, gynes, and males (cont’d) of M. mikromelanos sp. nov. and M. zeteki. Presented morphometrics were chosen based on vectors from NMDS plots with the most directionality (see Fig. 1A). All samples except for singleton workers of M. mikromelanos worker (Panama, Darién Provence) and gyne (Costa Rica, Heredia Provence) are from the Panama Canal Zone.

	Mean	SE	Min	Max	
M. mikromelanos sp. nov. workers
N = 117 (28 nests)					
HW	1.123	0.004	1.012	1.215	
EL	0.201	0.001	0.165	0.230	
ML	1.657	0.007	1.441	1.911	
PL	0.409	0.004	0.310	0.507	
CI	114.007	0.339	102.761	123.517	
SI	91.829	0.226	87.792	100.194	
TL	4.402	0.017	3.843	4.877	
M. zeteki workers
N = 54 (13 nests)					
HW	1.435	0.013	1.208	1.602	
EL	0.256	0.002	0.207	0.286	
ML	1.972	0.018	1.665	2.248	
PL	0.482	0.007	0.385	0.605	
CI	122.141	0.390	115.790	127.767	
SI	81.329	0.311	77.441	86.661	
TL	5.181	0.046	4.520	6.043	
M. mikromelanos sp. nov. gynes
N = 25 (9 nests)					
HW	1.298	0.005	1.267	1.380	
EL	0.270	0.004	0.231	0.304	
ML	2.028	0.008	1.915	2.112	
PL	0.599	0.010	0.507	0.704	
CI	113.715	0.535	104.546	119.481	
SI	79.847	0.405	73.478	84.451	
TL	5.659	0.018	5.435	5.814	
M. zeteki gynes
N = 28 (10 nests)					
HW	1.652	0.011	1.462	1.800	
EL	0.378	0.003	0.333	0.399	
ML	2.478	0.012	2.272	2.584	
PL	0.756	0.016	0.539	0.873	
CI	119.933	0.423	115.980	126.677	
SI	73.602	0.255	71.111	75.923	
TL	6.965	0.043	6.072	7.296	
M. mikromelanos sp. nov. males
N = 21 (5 nests)					
HW	0.840	0.004	0.817	0.893	
EL	0.294	0.003	0.282	0.310	
ML	1.781	0.012	1.671	1.859	
PL	0.369	0.006	0.338	0.437	
CI	122.934	0.956	115.437	134.084	
SI	98.407	0.618	91.713	103.427	
TL	4.550	0.035	4.196	4.759	
M. zeteki males
N = 22 (6 nests)					
HW	1.043	0.009	0.957	1.140	
EL	0.368	0.003	0.338	0.399	
ML	2.015	0.020	1.830	2.168	
PL	0.500	0.010	0.422	0.591	
CI	130.871	1.012	125.826	142.322	
SI	96.988	0.822	86.822	102.687	
TL	5.334	0.059	4.984	5.902	

Figure 1 Integrative taxonomy results.

Non-metric multidimensional scaling (NMDS) plots of worker (A), gyne or queen (B) and male (C) morphology. HW = head width, HL = head length, SL = scape length, EL = eye length; FL = frontal lobes, ML = mesosoma length, PL = petiole length, PPL = post-petiole length, GL = gaster length, CI = cephalic index, EI = eye index, SI = scape index, FLI = frontal lobe index, WI = waist length, TL = total length. SL is removed from worker NMDS (A) given its short vector and to show the M. mikromelanos sp. nov. type. Type specimen are indicated by a solid bold outline for both species, where the synonymized M. balboai (now M. zeteki; Weber 1958) is indicated with a dashed line, and paratypes are indicated by X’s. (D) Worker tempo differences between M. zeteki and M. mikromelanos sp. nov. (E) Reconstructed multi-locus phylogeny indicates that the published genome from (Nygaard et al., 2016; GCA_001594055.1) belongs to the new species, M. mikromelanos sp. nov.. Chemical chromatograms of M. mikromelanos sp. nov. (F) and M. zeteki (G) worker gasters indicating abundance differences of farnesenes: (1) E-β-farnesene, (2) (3Z,6E)-α-farnesene, (3) (3E,6E)-α-farnesene; with M. mikromelanos sp. nov. having a high abundance of (3) and low abundances (1) and (2), and M. zeteki having high a high abundance of (1) and low abundance (2) and (3).

MALES: For our male partition, all but FLI (p = 0.0307) were significantly different between species (Table 2). The NMDS converged on a two-dimensional solution with a robust stress level (stress = 0.1554). The Shepard plot also showed relatively high correlation with the regression line (non-metric fit R2 = 0.976; linear fit R2 = 0.886; see Fig. S1c). The NMDS plot showed M. mikromelanos sp. nov. and M. zeteki each forming distinct clusters with no outliers. The vectors for PL, mesosoma length (ML), SL, and cephalic index (CI) show the most strength in directionality of the measurements relative to the NMDS axes (Fig. 1C). The paratypes for both males fell well within their species clusters.

Our morphometric analysis shows that M. mikromelanos sp. nov. and M. zeteki are distinct species while supporting the previous synonymy of M. balboai under M. zeteki by Weber (1958b). Nearly all of the measurements taken are significantly different for all castes. The NMDS plots reflect the overlap of some measurements observed in workers and males while depicting clear separation of measurements observed in gynes.

Behavioral assay

The tempo of worker activity differed between the two species (Fig. 1D). A gamma distribution with an inverse link function was the best fit model (Table 3). For our diagnostic analysis of our GLMM see supplementary material (Figs. S2–S4). The gamma inverse model shows that tempo was correlated with species (Table 3, Pr (>—z—) = 1.150 ×10−02) and the variance of the random effect (colony) was not significant (var. = 7.977 ×10−02). This indicates that the variation observed in tempo was associated with species identity rather than with the particular colony of origin. This result provides further support for the delimitation between M. zeteki and M. mikromelanos sp. nov.

Phylogenetic analysis

Using published data (Nygaard et al., 2016; Solomon et al., 2019) located in GenBank (Mycetomoellerius zeteki genome: GCA_001594055.1) and the Mycetomoellerius gene sequences (Dryad DOI: 10.5061/dryad.2p7r771; GenBank accession numbers Table S2) we found genetic differences between M. mikromelanos sp. nov. and M. zeteki, with the former supported as genetically distinct from the latter by 100% bootstrap support (Fig. 1E). We located scaffolds for four genes (i.e., EF1α-F1, EF1α-F2, LwRh, and WG) and found high support for each in the published genome. For the mitochondrial gene COI, commonly used for DNA barcoding (Simon et al., 1994), 12 scaffolds were identified in the M. zeteki genome and only five had >95% query cover (Table S3) suggesting the presence of pseudogenes and rendering this marker unreliable (Leite, 2012). Based on the BIC scores, Modelfinder joined EF1α-F1 + WG (62 unique and 19 informative of 1777 sites) and EF1α-F2 + LwRh (13 unique and 6 informative of 972 sites) partitions and found the K2P+I and K2P to be the best fit models for those partitions respectively. The samples RMMA090930-09, RMMA050105-29, JSC030826-01, and the genomic scaffold sequences used (GCA_001594055.1) were identified as identical. Our phylogenetic analysis using four genes provided strong support for identifying the Nygaard et al. (2016) genome as belonging to M. mikromelanos sp. nov. rather than to M. zeteki as reported.

Table 3 Tempo analysis results.

Generalized linear mixed-effects models tested and compared. Each model had 40 observations, 20 each for M. mikromelanos sp. nov. and M. zeteki. Species and constant rows are the fixed effects estimates. Values in parentheses are standard errors (SE) for the cell above. The Gamma Inverse model was the best-fit model based on all diagnostics measures.

Fixed effects	Linear mixed model	Gaussian Log-Link	Gamma Log-Link	Gamma identity	Gamma inverse	
Species
(SE)	−0.593***
(0.161)	0.348***
(0.116)	0.343***
(0.101)	0.203***
(0.055)	−0.589 ***
(0.181)	
Constant
(SE)	2.031***
(0.132)	−0.712***
(0.090)	−0.709***
(0.072)	0.492***
(0.032)	2.035 ***
(0.143)	
Log Likelihood	8.887	12.889	14.765	14.711	14.848	
Akaike Inf. Criterion	−9.739	−17.778	−21.530	−21.422	−21.695	
Bayesian Inf. Criterion	−2.983	−11.023	−14.774	−14.667	−14.940	
Notes.

*** p < 0.01.

Chemical analysis

We found three farnesene compounds in M. mikromelanos sp. nov. and M. zeteki workers (1) E-β-farnesene, (2) (3Z,6E)-α-farnesene, and (3) (3E,6E)-α-farnesene, in whole samples and gaster trisections. Farnesenes have been reported before and are presumably localized in the gaster, functioning as trail pheromones (Adams et al., 2012a; Figs. 1F & 1G; Table 4). (3E,6E)-α-farnesene (3) is most abundant in M. mikromelanos, averaging 69.3% of the observed farnesenes. (1) and (2), are each at less than 23% of the overall abundance in M. mikromelanos sp. nov. E-β-farnesene (1), is the most abundant (62.2%) in M. zeteki with (2) at 18.4% and (3–5) with 6.5%.

These results illustrate that unique worker chemical profiles distinguish the two species. Some samples contained dilute concentrations of compounds as seen by the relative abundance (Figs. 1F & 1G). One M. mikromelanos sp. nov. colony (CRC170518-08) has a chemical profile similar to M. zeteki, with (1) 56.9%, (2) 33.7%, and (3) 9.3%. While this one colony stands out, all of the colonies of M. mikromelanos sp. nov. analyzed are morphologically distinct from M. zeteki and fit the description of M. mikromelanos (see Taxonomy section).

Table 4 Volatile worker compounds.

The average relative amount of six M. mikromelanos sp. nov. and four M. zeteki colonies sampled and standard error for compounds found in workers of both species. Worker farnesenes Farn. 1: E-β- farnesene, Farn. 2: (3Z,6E)-α-farnesene, and Farn. 3: (3E,6E)-α-farnesene.

Compound	Farn. 1	Farn. 2	Farn. 3	
Retention time		18.27	18.74	18.93	
mikromelanos sp. nov.	6	0.133
±0.08	0.089
±0.04	0.693
±0.13	
zeteki	4	0.622
±0.14	0.184
±0.10	0.065
±0.01	

Literature review

We found 63 articles that used M. zeteki or M. cf. zeteki under our search criteria (see Appendix Table S1). Twenty-eight articles did not identify the repositories of their voucher specimens, and of these, three articles deposited online sequence vouchers for ant specimens but mentioned no corresponding voucher specimens; nine others deposited symbiont vouchers (two fungal cultivar and seven non-cultivar symbionts). Voucher specimens were deposited in museums around the globe (Appendix Table S1), with the greatest number (fifteen) deposited at the Smithsonian Institution National Museum of Natural History, United States (USNM). The full list of voucher repositories includes: Colección Nacional de Referencia Museo de Invertebrados Universidad de Panamá (Panama); Smithsonian Tropical Research Institute Panama (Panama); Museu de Zoologia da Universidade de São Paulo (Brazil); Instituto Nacional de Biodiversidad (Costa Rica); Museo de Entomología de la Universidad del Valle (Colombia); Museo Entomológico Universidad Nacional Agronomía Bogotá (Colombia); Museum at the Universidad Técnica Particular de Loja (Ecuador); Natural History Museum of Denmark, (Denmark); Zoological Museum of the University of Copenhagen (Denmark); Zoological Museum, University of Puerto Rico (Puerto Rico); and the Smithsonian Institution National Museum of Natural History, (United States of America).

Mycetomoellerius mikromelanos sp. nov. Cardenas, Schultz, & Adams, new species	
Figs. 2 and 3 include M. mikromelanos.	

Geographic range: Panama: Colón, Darién, and Panama Province (RMMA and Jack Longino (JTL) specimens).

Label text: Separate labels for each specimen indicated by brackets (e.g., [Label 1] [Label 2]).

HOLOTYPE: Worker, Republic of Panama. [9.16328, -79.74413, Panama: Colón Province, Pipeline Rd, 16E, 62m, 13.v.2017, Cody Raul Cardenas, CRC170513-04] [USNMENT01123723]. Repository: USNM.

Figure 2 Mycetomoellerius mikromelanos sp. nov. type (A–C) and gyne paratype (D–F) specimens.

(A) Worker profile. (B) Worker head full-face view; FL = Frontal Lobe spine; CE = Compound Eye; Dsc = Head capsule Disc; Scp = Scape. (C) Dorsal worker view. (D) gyne paratype full-face view; AR = Arcuate Ridge; OS = Occipital Spine; Oc = Ocelli. (E) Gyne lateral view, Ha = Hamuli. (F) Gyne dorsal view Sclp = sculpturing.

Figure 3 Mycetomoellerius mikromelanos sp. n. male paratype specimen.

(A) profile, lateral view, Sclp = Sculpturing; PS = Propodeal Spine. (B) head full-face view Cly = Clypeus impression corners; OC = Oceli.

PARATYPES: 15 Workers, Republic of Panama. Same label data as holotype. Repositories: USNM (3): USNMENT01123726, USNMENT01123727, USNMENT01123728; MZSP (4): OSUC 640618, OSUC 640619, OSUC 640620, OSUC 640621; STRI (5): OSUC 640635, OSUC 640636, OSUC 640637, OSUC 640638, OSUC 640639; OSUC (3): OSUC 640606, OSUC 640607, OSUC 640608.

PARATYPES: 11 Gynes, Republic of Panama. Same label data as holotype. Repositories: USNM (4): USNMENT01123724, USNMENT01123729, USNMENT01123730, USNMENT01123731; MZSP (3): OSUC 640622, OSUC 640623, OSUC 640624; STRI (3): OSUC 640640, OSUC 640641, OSUC 640642; OSUC (1): OSUC 640609.

PARATYPES: 7 Males, Republic of Panama. Same label data as holotype. Repositories: USNM (4): USNMENT01123725, USNMENT01123732, USNMENT01129733, USNMENT01129734; MZSP (1): OSUC 640625; STRI (1): OSUC 640643; OSUC (1): OSUC 640610.

HOLOTYPE/PARATYPE Colony Code: CRC170513-04.

Additional material examined:

Workers N = 12: USNM: 3 specimens sharing label data [PANAMA: Pipeline RD, La Seda River; 79.736°′W 9.1529°′N; 28 v 2010;] [Henrick H. De Fine Licht; nest series; river bank; underground’ HDFL28052010-4 ch1][Trachymyrmex zeteki] [Check cryo] [DO NOT REMOVE SI DB Reference Not a property tag T. Schultz, NMNH] USNMENT00752565, USNMENT00752578, USNMENT00752579; 4 specimens sharing label data [PANAMA: Pipeline Rd, La Seda River; 79.736°W, 9.1529°′N; 28 v 2010;] [Henrik H. De Fine Licht nest series; river bank; underground HDFL28052010-5] [Trachymyrmex zeteki] [See cyro collections] [DO NOT REMOVE SI DB Reference Not a property tag T. Schultz, NMNH]: USNMENT00752574, USNMENT00752580, USNMENT00752581, USNMENT00752582; 3 specimens sharing label data [PANAMA: Pipeline Road, 2km past Limbo River 12v2010] [Henrik H. De Fine Licht; nest series; river bank; underground HDFL120502010-14] [Trachymyrmex zeteki] [See also cryo collections] [DO NOT REMOVE SI DB Reference Not a property tag T. Schultz, NMNH]: USNMENT00752565, USNMENT00752578 (1 pin with 2 specimens), USNMENT00752579. JTL: 1 specimen [PANAMA, Darién: 5 km S Platanilla 8.78105 -78-.41251 ±20 m 160 m, 20an2015 J. Longino#9082] [2nd growth veg. stream edge nest in clay bank] [CASENT0633645].

Males N = 3: USNM: 3 specimens sharing label data [PANAMA: Pipeline Road, 2km past Limbo River 12v2010] [Henrik H. De Fine Licht; nest series; riverbank; underground HDFL120502010-14] [Trachymyrmex zeteki] [See also cryo collections] [DO NOT REMOVE SI DB Reference Not a property tag T. Schultz, NMNH] USNMENT00752576, USNMENT00752578 (1 pin with 2 specimens).

Note: A name previously applied to this species, Trachymyrmex fovater, was incorrectly electronically published in a conference poster format and is therefore unavailable (Cardenas et al., 2016). This name is unavailable because (i) the date of the publication was not indicated on the poster and (ii) the name was not registered in the Official Register of Zoological Nomenclature (ICZN, 1999). We hereby describe Mycetomoellerius mikromelanos sp. nov. (LSID: urn:lsid:zoobank.org:act:B6BABA13-708F-44D8-AD2C-F4D5B8FB03E8), a name more appropriate for this species (see Etymology) and provide a complete diagnosis and description of this new species.

Diagnosis: Measurements for all castes are in Table 2 and Table S4. We found characters that reliably separate M. mikromelanos sp. nov. from M. zeteki. However, due to the variability of worker castes (e.g., mesosoma spines), intermediate character states occur in some individuals. The following characters are those most useful for diagnosis.

Workers (1) cuticle coloration dark-ferrugineous (Figs. 2A–2C); (2) overall integument bearing granulose irrorate sculpturing (Figs. 2A–2C); (3) frontal lobe with crenate margins and weak antero-lateral spine (Fig. 2B); (4) hooked spatulate bi-colored setae medial to frontal carina on disc of head capsule (Fig. 2B); (5) scape surpassing occipital corners when lodged in antennal scrobe (Fig. 2B); (6) convex margin of the compound eye extending past the lateral border of the head by more than half of its visible diameter in full-face view (Fig. 2B).

Gynes (1) cuticle coloration dark-ferrugineous (Figs. 2D–2F); (2) supraocular spine superior to compound eye by more than or equal to the eye length (Fig. 2D); (3) small arcuate ridge superior to and reaching anterior ocellus, with its terminal ends directed postero-laterally (Fig. 2D); (4) lateral ocelli partially obscured in full-face view (Fig. 2D); (5) mesoscutum with random-reticulate sculpturing (Fig. 2F & S5a); (6) wings bicolored, venation ferrugineous-brown (Figs. 2E, 2F] & S5b); (7) hindwing with 7–9 hamuli (Fig. 2E & S5b).

Males (1) bicolored; head and mesosoma ferrugineous-brown; metasoma dark testaceous-orange (Fig. 3A); (2) complete carinate-rugulose sculpturing of posterior head capsule, arranged nearly perpendicular to the longitudinal axis of the head (Fig. 3A); inferior to frontal lobe, sculpturing sparsely carinate and finely reticulate (Fig. 3A & S6); (3) mandible distinctly smaller compared to M. zeteki; (4) corners of medial clypeal emargination rounded (Fig. 3B & S6) ocelli smaller relative to M. zeteki in full-face view, occipital corner of head capsule visible (Fig. 3B); (6) propodeal spines wider at base than long (Fig. 3A).

WORKER: Overall pilosity is strongly bicolored, terminating with light coloration when spatulate; unless otherwise noted curved, appressed, and simple (Figs. 2A–2C). Older workers dark-ferrugineous; younger workers ferrugineous-orange. Integument typically with granulose irrorate sculpturing and a variably present white cuticular bacterial bloom.

Head: Disc of head capsule bears spatulate bi-colored setae (Fig. 2B); weakly granulose sculpturing; in full-face view broader than long. Mandible feebly sinuous, with 6–9 denticles. Palpal formula 4,2. Median margin of clypeus impressed; lateral-most corners of impression distinctly angulate. Preocular carina originate from mandibular insertion and terminate at occipital corners by a stout multituberculate tumulus directed postero-laterally. Frons with simple bi-colored setae. Frontal lobe semicircular; crenate margins and weak antero-lateral spine (Fig. 2B); frontal carina extending from posterior margins reaching occipital corners, joining the sub-parallel preocular carina to form antennal scrobes. Eye with 6-7 facets across width; eye margin extending past the lateral border of the head by more than half of its visible diameter in full-face view (Fig. 2B). Antenna with 11 segments; when lodged in antennal scrobe scape surpasses occipital corner (Fig. 2B); scape wide proximally and weakly tapering before thickening sub-distally; scape narrow at apex. Supraocular projection stout and multituberculate. Vertex impression shallow and narrow, but variable.

Mesosoma: Erect and strongly curved spatulate setae typically occurring from, or near, tubercles or spines; sparse rugulose sculpturing; most mesosomal sclerites with fine granulate sculpturing. Pronotum with fused median pronotal tubercles; superior lateral pronotal spine project antero-laterally; inferior lateral pronotal spines that project anteroventrally; in most cases the median pronotal spine projects as far or farther than lateral pronotal spine. Coxa I with entirely simple weakly bicolored setae; subtle superior impression on its anterior margin. Coxae II and III have spatulate setae on parallel carina dorso-laterally. Legs with spatulate setae proximally gradually becoming simple, appressed, and pale distally. Propodeum, in lateral view presents tuberculate carina at anterior base of propodeal spine; superior margin of metapleural gland bulla with variable number of tubercles; carina occurring from spiracle to propodeal lobes.

Metasoma: Petiolar nodes granulate and present a variable number of spines. Petiole with spatulate seta medially and posteriorly; intermittent carina comprised of tubercles; carina turn weakly mesad anteriorly but do not touch each other; lateral posterior margin weakly convex; in dorsal view anterior margins rounded (Fig. 2C); in dorsal view lateral margins subtly crenulate and weakly concave sub-anteriorally; ventral carinae converges to sub-petiolar process. Postpetiole with spatulate setae scattered dorsally and laterally; pair of simple setae ventrally; dorsal carina comprised of tubercles; in dorsal view, broader than long dorsally; posterior margin flat medially, with medial impressions on lateral margins. Gaster tergites and sternites with spatulate setae anteriorly; posterior margin of first tergite with subtly curved, simple setae; all other tergites and sternites with simple setae that become gradually finer and lighter posteriorly; weak reticulate sculpturing; triangular; in lateral view mostly round; first gastral tergite has crenate postero-lateral corners that surpass thin shiny margin between tergites I and II.

GYNE: Dark, curved, and simple setae unless otherwise noted; queens uniform ferrugineous-orange color; increasingly dark-ferrugineous with age. Integument generally with irrorate sculpturing and a variably present white cuticular bacterial bloom (Figs. 2D–2F).

Head: Setae of head capsule dark, curved, appressed, and simple; disc of head capsule bearing some spatulate setae; prominent sculpturing throughout; in full-face view, head longer than broad. Mandible feebly sinuous, with 6–8 denticles. Clypeus with minute tubercles scattered from anterior margin to slightly anterior of frontal lobes. Frontal lobe disc weakly rugulose; antero-lateral margin with reduced spine; semicircular; carina interior and parallel to margins. In full-face view, at least three quarters of the anterior lateral margin of compound eye surpassing lateral margin of head capsule; supraocular spine separated from compound eye by as much or more than the eye length (Fig. 2D; i.e., EL = 0.27 mm, distance to supraocular spine = 0.31 mm). Antennal scape wide proximally and tapering slightly before thickening sub-distally. Vertex carina extending from ocelli to frontal carinae; small arcuate ridge touches posterior margin of ocellus superior to anterior ocellus with terminal ends directed variably laterally and posteriorly but never anteriorly. Vertex variably impressed, but generally shallow and narrow.

Mesosoma: Sclerites with spatulate setae; confused-rugulose sculpturing. Pronotum with stout medial spine projecting anteriorly; superior lateral pronotal spine projecting antero-laterally; inferior lateral pronotal spine flattened laterally and projecting ventro-laterally. Coxa I with dark curved anterior setae and minute dense lightly colored pilosity throughout; weak asperous sculpturing on lateral face. Coxa II with bicolored weakly to fully spatulate setae along parallel carinae; a row of thick, dark, curved setae on posterior side in lateral view; rugulose sculpturing lateral to carinae. Coxa III has bicolored weakly to fully spatulate setae along carinae otherwise simple setae throughout; rugulose sculpturing lateral to carinae. Mesoscutum with appressed weakly spatulate to simple bicolored setae; random reticulate sculpturing. Mesoscutellar disc with appressed setae; random reticulate sculpturing (Fig. 2F & S5a); two small posteriorly projecting spines. Axilla hides scutoscutellar sulcus. Katepisternum and anepisternum suture embossed with strigate sculpturing. Inferior margin of anepisternum crenulate. Propodeal declivity nearly vertical.

Wings: Wings with a fine pubescence. Tegula with fine curved setae; triangular; weakly impressed on its face. Axillary sclerite with fine curved setae; flattened along distal margin. Forewing tinted smokey gray, more so anteriorly and less so posteriorly; venation ferrugineous-brown; with five cells (Fig. 2F & S5b); length of radial sector-media greater than half the length of the radius radial sector (Fig. S5b). Hindwing with long fine setae on posterior margin, longer proximally than distally; tinted smokey gray, more so anteriorly and less so posteriorly; venation ferrugineous-brown; two cells; 7–9 hamuli (Figs. 2D, 2F & S5b).

Metasoma: Petiole with weakly curved and bicolored setae, variably spatulate to simple; dorsal carinae of petiole with parallel spines that touch posterior margins; dorsal carinae directed medioanteriorly but not joining; ventral carinulae converging posteriorly on sub-petiolar process. Post-petiolar dorsum with distinct tubercles; lightly impressed medially; in dorsal view bearing two impressions on postero-lateral margins. Gaster with mostly simple setae, very few spatulate setae; terminal tergites have dense, lightly colored setae surrounded by dark setae; setae becoming less appressed towards terminal tergites and sternites; generally with strong confused reticulate sculpturing. First sternite and first tergite with confused-reticulate sculpturing; tergites I-IV have crenulate carinae bordering narrow shiny posterior margin.

MALE: Strongly appressed dark pilosity; mature males bicolored; head and mesosoma testaceous-orange and dark-ferrugineous, in part due to darkened sculpturing; metasoma testaceous-orange (Fig. 3A); integument with weak to effaced rugulose sculpturing (Fig. 3A).

Head: Pilosity dark and appressed to weakly appressed and curved; head capsule generally with carinate-rugose sculpturing; but sparsely carinate and finely reticulate inferior and lateral to frontal lobe; striate sculpture of head capsule in profile arranged nearly perpendicular to the longitudinal axis of the head (Fig. 3A; see also Fig. S6); head capsule in full-face view wider than long (Fig. 3B). Mandible with sparse, pale, and appressed setae; apical masticatory margin darker than rest of mandible; elongate-triangular and feebly sinuous; external margin feebly sinuate; prominent apical teeth; 4–6 mostly uniformly teeth. Clypeus absent to weakly sculptured; evenly rounded; narrow shiny anterior margin. Frons bulbous with weak to effaced carinate sculpturing across its entirety, forming two small mounds inferior to the frontal lobes. In lateral view, preocular carina occur near mandibular insertion, continue along inner margin of eye variably extending posterad. Frontal lobe with fine pilosity along margin but strongly curved setae on disc; strongly impressed medially; otherwise, smooth margin (Fig. 3B). Antennae covered with very fine lightly colored setae appressed (Fig. 3B); 13 segments; neck of scape and basal condyle visible (Fig. 3B); scape wide proximally, gently narrowing to apex. In full-face view lateral ocelli prominent and parallel to a shallow vertex impression (Fig. 3B). Supraocular projection absent or weak when, occurring directed posteriorly and near ocellus in full-face view.

Mesosoma: Setae strongly appressed throughout; sculpturing weak to effaced carinulate-rugulous, finely reticulate where carinulate-rugulous sculpturing absent. Pronotum with minute lateral spines projecting antero-laterally; median pronotal tubercle varying from clearly visible to greatly reduced, best seen laterally; forward-projecting median pronotal tubercle near mesoscutum and pronotal suture; inferior corner of pronotum, anterior to coxa I, carinae bear extremely reduced or absent inferior spine. Coxa mostly covered with light-colored setae. Coxa I with carinulate-rugulose sculpturing; coxa II with dark prominent setae posteriorly near trochanter; coxa I longer than coxa III, coxa II shortest. Mesoscutum, in lateral view, rounded and bulbous anteriorly, bulging over pronotal-mesoscutal suture. Mesoscutellar disc with two very small, posteriorly projecting spines. Propodeum with small posterior spines that are wider, or as wide at the base as long, projecting postero-laterally (Fig. 3A).

Wings: Overall pubescence fine. Forewing weakly bicolored; five cells; media-cubitus vein exceeds half-length of anal vein after the cubitus-anal vein proximally; length of radial sector-media greater than half the length of the radius radial sector . Hindwing with long fine setae on posterior margin, longer proximally than distally; uniform coloration, 6–8 hamuli.

Metasoma: Overall pilosity appressed; weakly sculptured; somewhat bicolored. Petiole with few curved and appressed setae dorsally; weakly costulate sculpturing; rounded; spiracle anterior to center; dorsally the lateral margins impressed, with anterior spine larger. Postpetiole with few curved and appressed setae dorsally; two ventral setae; in lateral view nearly rectangular; posterior margin shallowly impressed. Setae of first gastral tergite and sternite appressed; those on tergites 2–5, weakly appressed along posterior margins fine reticulate sculpturing. Pygostyle and genital opening densely covered with lightly colored setae.

Etymology

“Mikromelanos” is a singular, masculine adjective, compounded from the Greek μικρóς (mikrós), meaning “small,” and μελανóς (melanós), meaning “black” or “dark.” This etymology highlights the authors’ colloquial use of “little black” to describe the small darker queens of M. mikromelanos.

Comments

Although M. mikromelanos shares many similarities with M. zeteki (Figs. 2–5; Weber, 1940; Weber, 1958b; Mayhé-Nunes & Brandão, 2007), certain key characters allow us to easily distinguish the two species with a 20X loupe in the field. These key characters in M. mikromelanos are (i) the worker scapes extend past the occipital corners of the head capsule (extending only to the occipital corners in M. zeteki), (ii) gyne wing venation is ferrugineous-brown in M. mikromelanos and testaceous-orange in M. zeteki, (iii) queens of M. mikromelanos are typically smaller and a dark reddish brown, where M. zeteki queens are larger and a bright reddish color, (iv) males are bi-colored, dark-ferruginous and testaceous-orange (uniform, testaceous-orange in M. zeteki), and (v) in general, all castes of M. mikromelanos are smaller than those of M. zeteki. Distinguishing between the gynes or queens of M. mikromelanos and M. zeteki, however, requires a microscope. Aside from size, it is most informative to look at sculpturing of the mesoscutum under a microscope: M. mikromelanos gynes have random reticulate sculpturing on the mesoscutum whereas M. zeteki have parallel sculpturing. In addition to color differences, males of the two species can be differentiated by the integumental sculpture near the eye. In the male of M. mikromelanos ocelli are small and in lateral view the striations follow the contours of the ventroposterior borders of the eye (Fig. 3A & S6). Whereas M. zeteki ocelli are large and striations fan outward from the ventroposterior corner of the head and are interrupted by the borders of the eye and the preocular carina, where they end (Fig. 5A & S10). A complete list of measurements is provided in the Supplemental Information.

Biology

Mycetomoellerius mikromelanos is the most common ‘funnel Mycetomoellerius’ found on Pipeline Road, near Gamboa, Panama. Gynes establish nests from the start of the rainy season (May) into July. They nest in vertical clay embankments with entrances shaped like funnels (i.e., auricles) with flared margins (Mueller & Wcislo, 1998; Pérez-Ortega et al., 2010). Colonies are often tucked under roots or overhangs and occur in high densities (as close as ∼5cm apart) along creeks or are isolated in the forest at the base of trees. Colonies of M. mikromelanos have up to five vertically arranged chambers with single vertical tunnels between them. We removed the auricles from 16 nests and 15 were rebuilt to roughly the same size within seven days, suggesting the funnel structure appear to be biologically important (Figs. S7 & S8; also see Mueller & Wcislo, 1998; Schultz et al., 2002; Pérez-Ortega et al., 2010; Helms, Peeters & Fisher, 2014). Several functional hypotheses have been proposed from physical barriers for army-ant raids to visual nest recognition cues. But nest entrances could also be involved in gas exchange currents that disperse colony odors or assist with colony respiration (see Longino, 2005; Helms, Peeters & Fisher, 2014). Further research is necessary to clarify their biological function.

Figure 4 Mycetomoellerius zeteki worker (A–C) and gyne (D–F).

(A) Worker profile. (B) Worker head full-face view; FL = Frontal Lobe spine; CE = Compound Eye; Dsc = Head capsule Disc; Scp = Scape. (C) Dorsal worker view. (D) gyne paratype full-face view; AR = Arcuate Ridge; OS = Occipital Spine; Oc = Ocelli. (E) Gyne lateral view, Ha = Hamuli. (F) Gyne dorsal view Sclp = sculpturing.

Figure 5 Mycetomoellerius zeteki male.

(A) Profile, lateral view, Sclp = Sculpturing; PS = Propodeal Spine. (B) Head full-face view Cly = Clypeus impression corners; OC = Oceli.

A variety of organisms exploit the resources of M. mikromelanos (e.g., fungal garden, shelter, brood). Megalomyrmex adamsae (Longino, 2010), a rare obligate social parasite (1–6% parasitism rate), forages on the host garden and brood and never leaves the nest of M. mikromelanos (Adams et al., 2012b). Escovopsis Muchovej & Della Lucia, 1990, an assumed micro-filamentous fungal parasite, is maintained at low levels due to specialized grooming behaviors used by workers of M. mikromelanos (Currie, Mueller & Malloch, 1999; Currie et al., 2003; Little et al., 2003; Little et al., 2006). Other fungi such as Trichoderma (Persoon, 1794) threaten the health of the garden and are managed by the ants (Currie et al., 2003; Little et al., 2006). There are also six Diapriinae (Hymenoptera: Diapriidae) morphospecies exploiting M. mikromelanos, but little natural history has been reported for these associations (but see Pérez-Ortega et al., 2010). Diapriinae parasitoid wasps infiltrate nests and parasitize host larvae, turning them black as the wasps develop internally. We found that mature wasp pupae can be prompted to eclose when disturbed or picked up and male Acanthopria sp. Ashmead 1895 tend to naturally emerge before Acanthopria females in captive colonies (ca. 10 days; deposited in the RMMA collection). We also found that Mimopriella sp. (Masner & García, 2002) can take up to six months to complete development in a laboratory-maintained colony. The mechanism behind this unusually slow growth is unknown. These symbionts highlight the known diversity of a species network that is reliant on M. mikromelanos for survival.

Mycetomoellerius zeteki (Weber, 1940)	
Figs. 4 and 5 include M. zeteki.	

Geographic range: Colombia, Costa Rica, Ecuador, Panama (Mayhé-Nunes & Brandão, 2007)

Label text: Separate labels for each specimen indicated by brackets (e.g., [Label 1] [Label 2]).

LECTOTYPE (here designated): Worker; [Barro Colorado. CANAL ZONE No. 856 NAWeber 1938] [Trachymyrmex zeteki Weber COTYPE] [USNMENT01129855]. Repository: Museum of Comparative Zoology, United States of America (MCZ).

PARALECTOTYPE (here designated, examined): Worker; [Barro Colo. I. Canal Zone No.756 NAWeber 1938] [M.C.Z. CoType 25619] [T. zeteki Weber Cotypes] [Harbor Islands Insect Database] [MCZ-ENT 00025619]. Repository: MCZ.

Additional material examined

Workers N = 24: MCZ: 1 specimen with the label data [Barro Colo. I. Canal Zone No756 NAWeber 1938 walking at 9 pm. Snyder-Molino 0-4.] [762 1 worker USNM]; 1 pin with 2 specimens [Barro Colo. I. Canal Zone No. 759 NAWeber 1938] [T. balboai Weber Cotypes]. NHMB: 1 specimen with the label data [Barro Colo. I C.Z. 3441 NAWeber] [Trachymyrmex zeteki Weber] [17.vi.56 3441] [ANTWEB CASENT 0912534]; NOTE: The NHMB pin bears a “type” label, but we assume it to be erroneous because the specimen was collected in 1956 and therefore cannot be part of Weber’s 1938 M. zeteki syntype series. USNM: 3 specimens sharing these label data [PANAMA: Pipeline Rd; 19 v 2010; Henrik H. De Fine Licht; nest series; river bank; underground; HDFL1952010-8] [see also cyro collections] [Trachymyrmex sp’s] [DO NOT REMOVE SI DB Reference Not a property tag T. Schultz, NMNH] USNMENT00752570 (1 pin with 2 specimens), USNMENT00752572. 16 specimens sharing these label data: [9.1624,-79.74802, PANAMA: Colón, Pipeline Rd, Bird Plot 4E19N, 70m, 29.vi.2010, Rachelle M.M. Adams, RMMA100629-15] [Formicidae Myrmicinae Trachymyrmex zeteki, Weber, 1940, det. Cardenas, CR., 2018]. Repositories: USNM (4): USNMENT01129711, USNMENT01123714, USNMENT01123715, USNMENT01123716; MZSP (4): OSUC 640611, OSUC 640612, OSUC 640613, OSUC 640614; STRI (5): OSUC 640626, OSUC 640627, OSUC 640628, OSUC 640629, OSUC 640630; OSUC (3): OSUC 640601, OSUC 640602, OSUC 640603.

Gynes N = 9: Sharing these label data: [9.1624, -79.74802, PANAMA: Colón, Pipeline Rd, Bird Plot 4E19N, 70m, 29.vi.2010, Rachelle M.M. Adams, RMMA100629-15] [Formicidae Myrmicinae Trachymyrmex zeteki, Weber, 1940, det. Cardenas, CR., 2018]. Repositories: USNM (4): USNMENT01123712, USNMENT01123717, USNMENT01123718, USNMENT01123719. MZSP (2): OSUC 640615, OSUC 640616; STRI (2): OSUC 640633, OSUC 640634; OSUC (1) OSUC 640604.

Males N = 11: USNM: 3 specimens sharing these label data [PANAMA: Pipeline Rd; 19 v 2010; Henrik H. De Fine Licht; nest series; riverbank; underground; HDFL1952010-8] [see also cyro collections] [Trachymyrmex sp’s] [DO NOT REMOVE SI DB Reference Not a property tag T. Schultz, NMNH] USNMENT00752568 and USNMENT00752570 (1 pin with 2 specimens). Sharing these label data: [9.1624, -79.74802, PANAMA: Colón, Pipeline Rd, Bird Plot 4E19N, 70m, 29.vi.2010, Rachelle M.M. Adams, RMMA100629-15] [Formicidae Myrmicinae Trachymyrmex zeteki, Weber, 1940, det. Cardenas, CR., 2018]. Repositories: USNM (4): USNMENT01123713; USNMENT01123720; USNMENT01123721; USNMENT01123722; MZSP (1): OSUC 640617; STRI (2): OSUC 640633, OSUC 640634; OSUC (1): OSUC 640605.

Mycetomoellerius zeteki was originally described by Weber (1940) as Trachymyrmex zeteki from an accidental collection in dense shade on a slope near the lab on Barro Colorado Island, Panama Canal Zone (Weber, 1940; Mayhé-Nunes & Brandão, 2007). In the same article Weber followed his description of T. zeteki with a description of T. balboai (Weber, 1940). These descriptions were based on a small series of workers from single collections. Weber noted similarities between the two species in his original descriptions. According to Weber, T. zeteki was distinctly smaller than T. balboai, paler in appearance, and the relative proportions of the thoracic spines differed. The character states that Weber used to differentiate the two species were later understood to represent variation within a single species and T. balboai was synonymized with M. zeteki (Weber, 1958b). In Mayhé-Nunes & Brandão (2007) revision of the Trachymyrmex “Jamaicensis group,” M. zeteki was placed in a subset of the “Iheringi group.” Distinct characteristics of the Jamaicensis group are the open antennal scrobes arising from the subparallel preocular and frontal carinae (Mayhé-Nunes & Brandão, 2007), a character cited by Solomon et al. (2019) as applying to the entire genus Mycetomoellerius. Here we describe the gyne wing venation and males of M. zeteki and provide comparative morphology in the comments to delineate M. zeteki from M. mikromelanos. For complete descriptions of worker and gynes of M. zeteki, see Weber (1940); Weber (1958b) and Mayhé-Nunes & Brandão (2007).

Diagnosis: Measurements for all castes are found in Table 2, Table S4. Certain characters are useful for separating M. zeteki from M. mikromelanos sp. nov. However, due to the variability of the worker castes (e.g., mesosoma spines), intermediate character states occur in some individuals. The following characters are most useful.

Workers (1) cuticle ferrugineous (Figs. 4A–4C; dark-ferrugineous in M. mikromelanos); (2) integumental sculpture weakly irrorate (Figs. 4A & 4B; granulose irrorate sculpturing in M. mikromelanos); (3) frontal lobe with weakly crenulate margins and distinct antero-lateral spine (Fig. 4B; crenulations present and spines lacking in M. mikromelanos); (4) disc of head capsule between frontal carinae mostly lacking strongly hooked spatulate bi-colored setae (Fig. 4B; present in M. mikromelanos); (5) scape of antenna reaching occipital corners when lodged in antennal scrobe (Fig. 4B; surpassing occipital corners in M. mikromelanos); (6) convex margin of the compound eye extending past lateral border of head capsule by less than half of the eye area in full-face view (Fig. 4B; extending by more than half in M. mikromelanos).

Gyne (1) cuticle coloration ferrugineous (Figs. 4D–4F; dark-ferrugineous in M. mikromelanos); (2) supraocular tubercle separated from compound eye by a distance less than or equal to the eye length (Fig. 4D; more than or equal to eye length in M. mikromelanos); (3) small arcuate ridge superior to anterior ocellus with terminal ends directed antero-laterally (Fig. 4D; directed postero-laterally in M. mikromelanos); (4) lateral ocelli conspicuous in full-face view (Fig. 4D; partially obscured in M. mikromelanos); (5) mesosoma with sparse carinate sculpturing; mesoscutum with parallel-costulate sculpturing (Fig. 4F; random-reticulate in M. mikromelanos); (6) wing venation testaceous-orange brown (Figs. 4E & 4F; wings weakly ferrugineous-brown in M. mikromelanos); (7) hindwing with 5–8 hamuli (Fig. 4E; 7–9 in M. mikromelanos).

Male (1) coloration mostly uniform testaceous-orange (Fig. 5A; bicolored, head and mesosoma ferrugineous-brown with metasoma dark testaceous-orange in M. mikromelanos); (2) striations on head capsule fanning outward from ventroposterior corner of head, ending at the compound eye and preocular carina (Fig. 5A & S10; striations perpendicular to longitudinal axis in M. mikromelanos); sculpture prominent on posterior head capsule, minute to absent anteriorly (Fig. 5A; nearly complete sculpturing of head capsule in M. mikromelanos); (3) mandible larger compared to those of M. mikromelanos; (4) corners of clypeal emargination slightly angled (Fig. 5B; rounded in M. mikromelanos); (5) in full-face view; occipital corners of head capsule partially obscured by large ocelli (Fig. 5B; visible in M. mikromelanos); (6) propodeal spines longer than width of spine at base (Fig. 5A; wider at base than long in M. mikromelanos).

GYNE:

Wings: Overall pubescence fine. Tegula with dark appressed simple setae; testaceous-orange coloration; triangular; impressed face. Axillary sclerite with setae along ventral margins and dark appressed setae on its face; flattened along distal margin. Forewing weakly tinted smokey grey, only slightly more so anteriorly than posteriorly; venation testaceous-orange/brown (Fig. 4F); five cells; length of radial sector-media vein less than half the length of radius-radial sector vein. Hindwing with long fine setae on posterior margin, longer proximally than distally; lightly tinted smokey grey, venation testaceous-orange/brown; 5-8 hamuli (Figs. 4E & 4F).

MALE: Dark and weakly appressed setae; mature males nearly uniform testaceous-orange color; darkened sculpturing on head capsule, otherwise integument generally weak to effaced carinnulate-rugalose sculpturing (Fig. 5A).

Head: Poorly appressed curved dark setae; feebly darker than rest of body due to sculpturing; sculpturing carinulate-rugulose lateral and posterior to frontal lobes; sculpturing reduced posterior to median ocelli and in median portion of vertex; otherwise finely reticulate; striation on head capsule fanning outward from ventroposterior corner of head, ending at the compound eye and preocular carina (Figs. 5A & 5B; see also S10); head capsule in full-face view wider than long (Fig. 5B). Mandible with fine and lightly colored setae; setae on external margin appressed; masticatory margin distinctly darker than rest of mandible; triangular and feebly sinuous; apical tooth prominent; with 5–7 dentate to denticulate teeth. Clypeal margin somewhat shiny; not evenly rounded; weak angle near clypeal emargination. Frons with carinulate-ruglose sculpturing forming two small mounds bearing curved setae inferior to frontal lobes; otherwise mostly smooth; somewhat bulbous. Preocular carina originating near mandibular insertion, continuing along inner margin of eye and extending variably posterad. Frontal lobe with fine pilosity along margin but dark simple setae on disc; lightly impressed medially; otherwise, smooth margin. Antenna with 13 segments; scape covered in fine and intermittent dark appressed setae; scape and basal condyle visible; scape wide proximally and gently tapering before widening sub-distally to apex (Fig. 5B). In full-face view ocelli large and distinct; central and lateral ocelli are prominent, large, and distinct; lateral ocelli parallel to the shallow vertex impression. Supraocular projection directed posteriorly, visible in full-face view.

Mesosoma: Somewhat appressed setae throughout; carinulate-rugulous sculpturing; weakly reticulate when carinulate-rugulous sculpturing absent. Pronotum with small lateral spine projecting anteriorly; minute to entirely absent spine occurs medially along anepisternum pronotal suture; inferior corner, near coxa I, with extremely reduced spine. Coxa covered mostly with lightly colored setae; weak carinulate sculpture. Coxa I with a few dark setae anteriorly. Coxa II with dark prominent setae mostly ventrally near trochanter. Length of coxa III equal to or longer than coxa I. Mesoscutum, in lateral view, rounded and bulbous anteriorly, bulging over pronotal-mesoscutal suture. Axilla hide part of scutoscutellar suture in lateral view. Mesoscutellar disc with two very small, posteriorly projecting spines (Fig. 5A). Propodeal spines as long or longer than width of base and projecting posteriorly (Fig. 5A).

Wings: Overall pubescence fine. Forewing weakly bicolored; possessing five cell; media–cubitus vein less than half length of anal vein after cubitus-anal proximally; length of radius-radial sector vein less than half the length of radial sector-media. Hindwing with long fine setae on posterior margin, more so proximally than distally; uniform in coloration; 4–7 hamuli.

Metasoma: Mostly weakly appressed simple and subtly curved setae; uniform coloration; poorly sculptured. Petiole with curved setae dorsally; costulate sculpturing if present; node rounded; in profile spiracle present medially at anterior margin; in dorsal view antero-lateral tumuli flanking a flattened medial projection. Postpetiole sculpturing finely reticulate if present; posterior ventral side with spine present varying in length from absent to almost as long as postpetiole; in lateral view nearly square; shallow posterior impression. Gaster setae sparse and appressed; sculpturing finely reticulate; sternites and tergite setae sparse and appressed; sternite and tergite 2–5 with dark setae along posterior margins. Pygostyle and genital opening covered with lightly colored setae.

Comments

A specimen of M. zeteki deposited at the Natural History Museum, Basel Switzerland bears a “cotype” (i.e., syntype) label in error. The data label reads as follows ‘[Barro Colo. I C.Z. 3441 NAWeber] [17.vi.56 3441] [Trachymyrmex zeteki Weber] [ANTWEBCASENT0912534] [type].’ It is not possible that this specimen, collected in 1956, 18 years after the M. zeteki type series was collected, is a type specimen of that species. While this specimen could be part of the material examined in Weber, 1958b balboai-zeteki synonymy, no repositories were mentioned. This specimen was not treated as a syntype for this study. For a complete description of the workers and gyne of M. zeteki, see Mayhé-Nunes & Brandão (2007). Certain key characters allow us to easily distinguish M. zeteki from M. mikromelanos with a 20X loupe in the field. For M. zeteki these characters are (i) in workers of M. zeteki, the scapes reach the occipital corners of the head capsule but do not extend past them, whereas in M. mikromelanos, they extend past the head capsule when lodged in the antennal scrobe, (ii) the queens of M. zeteki are comparatively larger than those of M. mikromelanos and are typically bright reddish in color whereas M. mikromelanos are generally a darker reddish brown, (iii) gyne wing venation is testaceous-orange in M. zeteki and ferrugineous-brown in M. mikromelanos, (iv) males are uniform in color and testaceous-orange (bicolored dark-ferrugineous and testaceous-orange in M. mikromelanos), and (v) in general all castes of M. zeteki are larger than M. mikromelanos. It is necessary to note that workers from incipient colonies of M. zeteki tend to resemble workers of M. mikromelanos and require careful attention to the additional variable characters. A complete list of measurements can be found in the Supplemental Information.

Biology

Most reports of M. zeteki are most likely accounts of M. mikromelanos (Appendix Table S1). Mycetomoellerius zeteki is rare relative to M. mikromelanos in the Canal Zone near Gamboa, Panama. For example, we only located two colonies of M. zeteki near the type locality on Barro Colorado Island, and one colony at El Llano ca. 40 km east of the canal. On the mainland we have found mixed sites of both species and a single creek with only M. zeteki present (Rio Mendoza, ca. 1 km North of Rio La Seda), but when the two species occur together, M. zeteki always occurs at comparably lower densities. While some M. zeteki samples are indicated to be collected from Nicaragua we examined seven specimens from JTL and three specimens were morphologically distinct from the Panamanian M. zeteki and M. mikromelanos. Mycetomoellerius zeteki and M. mikromelanos are similar morphologically and biologically and this has led to confusion between these sister species. In both species, gynes establish their nests from the start of the rainy season (around May) into July. Nests can be found on the same clay embankments with indistinguishable auricles with up to five chambers. In the five mature M. zeteki nests we excavated, each had two tunnels connecting each chamber. In contrast M. mikromelanos has only one tunnel connecting them and there are likely other architectural differences, such as volume and internal auricle shape, but more colonies of M. zeteki need to be examined.

Mycetomoellerius zeteki and M. mikromelanos also have a similar range of symbionts. Megalomyrmex adamsae associates with M. zeteki, foraging on host garden and brood, and never leaves the host nest (Adams et al., 2012b). An Escovopsis fungal parasite attacks the fungal garden. Garden maintenance behavior also appears similar as M. zeteki forms infrabuccal pellet piles like M. mikromelanos (Little et al., 2003). We have documented the first Diapriidae wasp parasitizing the brood of M. zeteki. In a laboratory colony (CRC170519-01), we observed a male wasp of Mimopriella sp. (deposited in the RMMA collection) emerge on May 19th, 2017, and a female 10 days later. The live colony had characteristically black larvae when collected. While some natural history has been documented, there is still much more to be discovered about the symbionts, nest architecture, and general biology of M. zeteki.

Discussion

Based on multiple lines of evidence, we have shown that the new species M. mikromelanos is a well-studied cryptic species that has been confused with M. zeteki for decades. We accomplished this by examining morphology and morphometrics of all castes, analyzing the behavior of workers, comparing worker volatile compounds, and comparing DNA sequence data. Interestingly, we also determined that the published genome (Nygaard et al., 2016) belongs to the newly described species M. mikromelanos. Our results underscore the importance of species discovery by emphasizing the value of an integrative taxonomic approach, the effect of species delineation on biodiversity, and the necessity of properly vouchered specimens.

While historical taxonomic work generally relied on morphological characters alone to delineate and typify species, modern taxonomy more often utilizes other biological evidence (Dayrat, 2005; Schlick-Steiner et al., 2010). An integrative approach is frequently used to overcome the challenges of cryptic species, especially those lacking clear morphological characters adequate for recognizing species boundaries. Complementary lines of evidence in addition to morphology (e.g., behavioral, molecular, chemical, ecological, etc.) increase our confidence in species descriptions and reveal the intricacies of those species’ biology (Dayrat, 2005). Employing this approach, we analyzed biologically relevant evidence along with key morphological characters—summarized in the diagnoses of M. mikromelanos and M. zeteki—that proved useful for distinguishing the two species. These are best observed using a standard dissection microscope but can also be detected with a 20X loupe. Another line of evidence is provided by our behavioral analysis. It was initially assumed that tempo would reflect behavioral differences observed in the field, where M. zeteki appeared ‘aggressive’ and M. mikromelanos ‘passive’. However, we found that these two sibling species show differences in tempo, the rate of movement, rather than in aggressive or passive behaviors. Lastly, our chemical analysis also shows species-specific differences in the abundance of volatile compounds for most analyzed workers of both species. The combined evidence supports the existence of two distinct and closely related sympatric species in the Panama Canal Zone, M. mikromelanos and M. zeteki. The recognition of two species adds to our understanding of the multiple symbiotic relationships involving each species. It should be noted that, although it appears fairly certain that M. mikromelanos represents a single, well-supported species (Fig. 1E), the possibility remains that M. zeteki as currently defined may actually consist of two or more cryptic species. In Fig. 1E, all the samples of M. mikromelanos form a very well-supported clade whereas the monophyly of the two M. zeteki samples is poorly supported. This is also reflected in a larger phylogeny where the same two M. zeteki samples are monophyletic but have similarly poor support and long branch lengths (see Fig. 2 of Solomon et al., 2019).

Species delimitation is essential not only for descriptive biology, but also for understanding the levels of biodiversity. In this context, species represent units of study that help us comprehend ecological and evolutionary principals. These include, but are not limited to, genetic diversity, adaptation, and broad-scale community interactions. Fungus-growing ants are an intriguing group for the study of biodiversity given their coevolutionary history with their fungal cultivars (Mehdiabadi et al., 2012), their many other symbiotic relationships (Mueller, Rehner & Schultz, 1998; Currie, Mueller & Malloch, 1999; De Fine Licht & Boomsma, 2014), and the role fungus-growing ants play as ecosystem engineers (Jones, Lawton & Shachak, 1994; Folgarait, 1998; Meyer et al., 2011; Meyer et al., 2013). However, the distributions and ecological roles of most non-leaf-cutting attines in neotropical environments is still poorly studied (but see Leal & Oliveira, 2000; Vasconcelos, Araújo & Mayhé-Nunes, 2008; Tschinkel & Seal, 2016). For example, during the summer of 2018 we searched BCI, Fort Sherman, and El Llano (ca. 15, 35, and 80 km from Pipeline Road, respectively) for both M. mikromelanos and M. zeteki. Yet after searching kilometers of trails and creeks on BCI we were unable to locate any M. mikromelanos colonies, and only located two M. zeteki colonies on BCI and one at El Llano. No M. mikromelanos were found outside of the regularly sampled Gamboa Forest and Pipeline Road areas with the exception of a sample collected by Dr. Jack Longino in the Darien Provence of Panama in 2015. Regardless of our uncertainty of M. mikromelanos’ distribution outside of the Canal Zone, we do have some familiarity with M. mikromelanos’ and M. zeteki’s symbiotic associations. For example, they maintain similar relationships with social parasites, garden pathogens, and parasitoids (see “Biology” in species descriptions). Describing M. mikromelanos has enhanced our understanding of the symbiotic relationships of both species and raises more questions about them and their associates. Further research clarifying the natural history of these species and their symbionts will help us discern their ecological roles and contribute to our understanding of biodiversity in the Panama Canal Zone.

Genetic patterns and genetic diversity are another important aspect of biodiversity. Together they can inform understanding of the dispersal capabilities of species (Sanetra & Crozier, 2003; Sanllorente, Ruano & Tinaut, 2015; Boulay et al., 2017; Helms IV, 2018) biogeographic histories (Branstetter et al., 2017; Mueller et al., 2017; Mueller et al., 2018), demographic history (Castilla et al., 2016), and evolutionary patterns (Baer & Boomsma, 2004; Schultz & Brady, 2008; Nygaard et al., 2016; Mueller et al., 2018). Modern molecular genetic tools enable researchers to study populations and their patterns at broad biogeographic ranges. According to several studies, higher attine ants are believed to grow two clades of fungi (Solomon et al., 2019; Schultz et al., 2015; Ješovnik & Schultz, 2017; Shik et al., 2021; Mueller et al., 2018). Still, there is not a one-to-one association or complete phylogenetic congruence between higher attine ants and their cultivars (Schultz et al., 2015; Mueller et al., 2018). Including a broader sampling of species revealed that a lower attine species (e.g., Apterostigma megacephala (Lattke, 1999)) also cultivates a L. gongylophorus higher attine garden (Schultz et al., 2015). Furthermore, multiple cultivar strains or haplotypes can be cultivated by the same or different ant species from the same location (Green, Mueller & Adams, 2002; Shik et al., 2021). As in most scientific endeavors, new knowledge of ant-fungus associations requires constant updating of older models (Chapela et al., 1994; Mueller & Wcislo, 1998; Schultz & Brady, 2008; Mehdiabadi & Schultz, 2010). This process generates a deeper and more complicated picture of the biogeographic patterns observed in populations of the higher attines. Well-designed population-level analyses of the 61 non-leaf-cutting higher attine ant species (e.g., Mycetomoellerius, Paratrachymyrmex, Trachymyrmex, Xerolitor, and Sericomyrmex) would further refine our understanding of coevolution in the fungus-growing ants. Mycetomoellerius mikromelanos is well suited for such population-genetic analyses for a few reasons: it is abundant in the Canal Zone and easily located given its characteristic auricle nest entrance, it is sympatric with its sister species M. zeteki, and it has a published genome (Nygaard et al., 2016). Originally named Trachymyrmex zeteki on GenBank ((Nygaard et al., 2016); GenBank accession: GCA_001594055.1), we confirm in this study based on published nuclear gene sequences (see phylogenetic analysis) and morphological evidence of vouchers (see taxonomy; Figs. S11 & S12) that it is the genome of M. mikromelanos.

The published genome of M. mikromelanos highlights the importance of species identification and voucher specimen deposition. Physical vouchers provide reproducibility and confidence in published findings. Curating physical collections, naming species, and creating molecular databases still depend on non-molecular taxonomic work (Dayrat, 2005; Turney et al., 2015). We found that the incidence of reported vouchering for M. zeteki or M. cf. zeteki, based on our literature review, is higher than what is typically found in the field of entomology (44% versus 35%: Turney et al., 2015). This could be due to the exponential increase in research focusing on attines and collaborations with skilled taxonomists over the past thirty to forty years. We argue that more effort in voucher deposition is needed and that this is especially true when genomic information is published. Genomic resources are frequently used to compare and characterize gene functions (e.g., Lee et al., 2017; Nolasco et al., 2018; Wang et al., 2019). Incomplete taxonomic information can lead to a series of misguided future studies.

Conclusions

Given the abundance of M. mikromelanos in the Panama Canal Zone, we expect that the majority of researchers who believe they have studied M. zeteki have studied M. mikromelanos instead (Appendix Table S1). We encourage these researchers to mount specimens, confirm the species identification, and deposit the vouchers in a well-curated and accessible natural history museum collection. Our hope is that our results will encourage voucher deposition, even for common species such as M. mikromelanos. While physical voucher specimens are not typically required by journal policy or by reviewers (Turney et al., 2015), our findings draw attention to why this is important. We recommend that investigators include voucher specimen preparation and deposition as part of their normal research practice and instill this principle in mentees and colleagues.

Supplemental Information

Supplemental Information 1 Published research on Mycetomoellerius zeteki and M. cf.zeteki.

In some studies, it was assumed that the smaller black species was M. zeteki sensu stricto (e.g., Adams et al., 2012a; Adams et al., 2012b) whereas in others uncertainty was indicated by referring to the smaller, darker species as M. cf. zeteki. Vouchers that are not represented by a physical ant specimens are indicated by (*) for GenBank depositions, (†) for fungal-cultivar GenBank deposition, and (‡) for other symbiont GenBank depositions

Click here for additional data file.

Supplemental Information 2 Fungus-growing ant species described since from 1995 until July 2019

(*) Indicates new genera.

Click here for additional data file.

Supplemental Information 3 GenBank accession numbers for samples used in Solomon et al. (2019)

Click here for additional data file.

Supplemental Information 4 M. zeteki published genome

Scaffolds found when the consensus sequences were blasted to the published M. zeteki genome.

Click here for additional data file.

Supplemental Information 5 Partitioned morphometric measurements

Table of partitioned measurements for examined Mycetomoellerius mikromelanos sp. nov. (mikro.) and M. zeteki (zeteki) specimen by caste and species, including standard errors (SE), minimum and maximum standard measurements in millimeters, except for indices. The junior synonym of M. zeteki is included as “balboai” which are included in the entire M. zeteki partition.

Click here for additional data file.

Supplemental Information 6 NMDS stress plots for each caste

Click here for additional data file.

Supplemental Information 7 QQ-plot of gamma inverse model showing relatively normal quantiles

Click here for additional data file.

Supplemental Information 8 Fitted GLMM residuals indicating separation

Click here for additional data file.

Supplemental Information 9 Density plot with a nearly normal distribution

Click here for additional data file.

Supplemental Information 10 Mycetomoellerius mikromelanos sp. n. gyne mesoscutum and wings

(a) Dorsal view of mesoscutum presenting random-reticulate sculpturing. (b) Wing veination: Ha = hamuli, r-rs = radius-radial sector; rs-m = radial sector-media.

Click here for additional data file.

Supplemental Information 11 Mycetomoellerius mikromelanos sp. nov. male

Lateral view providing an additional view of the the head capsule sculpturing.

Click here for additional data file.

Supplemental Information 12 Measurements of height, width, and area taken from auricles.

Click here for additional data file.

Supplemental Information 13 Results of auricule knockdown experiment

(A) Width = w, height = h, length = l, (B) area = a, (c) shape = s; measurements post knockdown are indicated with a 2 (e.g., w = pre-knockdown, w2 = post knockdown).

Click here for additional data file.

Supplemental Information 14 Mycetomoellerius zeteki gyne mesoscutum and wings

(A) Dorsal view of mesoscutum presenting random-reticulate sculpturing. (B) Wing veination: Ha = hamuli, r-rs = radius-radial sector; rs-m = radial sector-media.

Click here for additional data file.

Supplemental Information 15 Mycetomoellerius zeteki male

Lateral view providing an additional view of the the head capsule sculpturing. Notable as well is the variable ventral post-petiol process observed on some M. zeteki male specimens.

Click here for additional data file.

Supplemental Information 16 Profile view of Mycetomoellerius mikromelanos sp. nov. genomic voucher specimen (Nygaard et al., 2016); GenBank accession: GCA_001594055.1)

Click here for additional data file.

Supplemental Information 17 Face view of Mycetomoellerius mikromelanos sp. nov. genomic voucher specimen (Nygaard et al., 2016); GenBank accession: GCA_001594055.1)

Click here for additional data file.

Supplemental Information 18 Supplemental Auricle Text

Click here for additional data file.

Supplemental Information 19 Supplemental references

Click here for additional data file.

Supplemental Information 20 Morphometric measurments and indices of workers, gynes, and males

Click here for additional data file.

Supplemental Information 21 Morphometric analysis R code

Click here for additional data file.

Supplemental Information 22 Behavioral analysis data

Click here for additional data file.

Supplemental Information 23 Generalized Linear Mixed Model R code

Generalized linear mixed model R analysis for tempo analysis

Click here for additional data file.

Supplemental Information 24 Concatenated molecular data

Data from Solmon et al. 2019 Systematic Entomology & published Trachymyrmex genome. Samples were mapped to reference in Geneious

Click here for additional data file.

Supplemental Information 25 IQ-TREE ModelFinder analysis

Click here for additional data file.

Supplemental Information 26 IQ-TREE phylogenetic analysis best scheme results

Click here for additional data file.

Supplemental Information 27 Retention time of worker volatile compounds

Click here for additional data file.

Supplemental Information 28 Auricle knockdown data

Click here for additional data file.

Supplemental Information 29 Specimens examined repositories

All specimens indicated in the manuscript with the associated specimen code, collection code, type status, and repository.

Click here for additional data file.

We thank the staff and researchers at the Smithsonian Tropical Research Institute (STRI) for logistical support and the Autoridad Nacional del Ambiente y el Mar for permission to sample and export ants. Some live colonies were provided by our colleague Morten Schiøtt, along with Matt F. Fisher and Konstantinos Giampoudakis in conjunction with the graduate course Tropical Behavioral Ecology and Evolution (TBEE) at STRI, hosted by the Centre for Social Evolution, University of Copenhagen and STRI in 2011 and by The Ohio State University and STRI in 2017 and 2019. Specimens were generously loaned from the Museum of Comparative Zoology, Harvard, Cambridge, Massachusetts, the National Museum of Natural History, Washington, D.C., The Natural History Museum Basel, Basel, Switzerland, and Jack Longino. We thank Panagiotis Sapountzis for assistance with the etymology, Luciana Musetti and Sarah Hemly from the Triplehorn Insect Collection, and David Culver, Joan M. Herbers, and Steven Passoa for microscopy support. We thank Alexander Wild for assistance with establishing an Adams Mega. Lab automontage system. We are grateful to Christopher Wilson, Marymegan Daly, and the Adams Mega Ant Lab peers for improving this work with extensive editing, conversation, and encouragement. We thank the reviewers Pepijn Kooij, Rodrigo Feitosa, and Monica Ulyssea and the editor Joeseph Gillespie for helpful comments on the final draft of the manuscript. Lastly, we dedicate this work to the late Christopher Wilson, who will be missed.

Additional Information and Declarations

Competing Interests

Author Contributions

Ethics

Field Study Permissions

Data Availability

New Species Registration

The authors declare there are no competing interests.

Cody Raul Cardenas conceived and designed the experiments, performed the experiments, analyzed the data, prepared figures and/or tables, authored or reviewed drafts of the paper, generated R code for morphometric and behavioral analysis, and approved the final draft.

Amy Rongyan Luo conceived and designed the experiments, performed the experiments, analyzed the data, prepared figures and/or tables, authored or reviewed drafts of the paper, generated R code behavioral analysis, and approved the final draft.

Tappey H. Jones performed the experiments, analyzed the data, authored or reviewed drafts of the paper, and approved the final draft.

Ted R. Schultz analyzed the data, authored or reviewed drafts of the paper, and approved the final draft.

Rachelle M.M. Adams conceived and designed the experiments, performed the experiments, analyzed the data, prepared figures and/or tables, authored or reviewed drafts of the paper, and approved the final draft.

The following information was supplied relating to ethical approvals (i.e., approving body and any reference numbers):

The Smithsonian Tropical Research Institute approved the project named ”Behavioral Ecology and Systematics of the Fungus-growing Ants and Their Symbionts (#4056)” to be conducted in Panama in its nature preserves.

The following information was supplied relating to field study approvals (i.e., approving body and any reference numbers):

Colony collection and field work were approved by the Smithsonian Tropical Research Institute and the Autoridad Nacional del Ambiente y el Mar (Permiso de Colecta Científica 2017: SPO-17-173 and 2018: SE/AB-1-18).

The United States Department of Agriculture approved the import of specimens and live colonies maintained at an approved facility (permit: APHIS P526P-16-02785; facility #4036).

The following information was supplied regarding data availability:

Raw data and associated R code for morphometrics, behavior, phylogenetic, and chemical analyses are available in the Supplemental Files and at GitHub: https://github.com/crcardenas/Mycetomoellerius_spn.

All specimens examined and the associated repositories are in the Supplemental File.

The following information was supplied regarding the registration of a newly described species:

Publication LSID: urn:lsid:zoobank.org:pub:737E04E5-5A8F-48F6-BE32-ADC1028927B6

Mycetomoellerius mikromelanos sp. n. LSID urn:lsid:zoobank.org:act:B6BABA13-708F-44D8-AD2C-F4D5B8FB03E8.

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
