# Peer review of "Using an integrative taxonomic approach to delimit a sibling species, Mycetomoellerius mikromelanos sp. nov. (Formicidae: Attini: Attina)"

_PeerJ, doi:10.7717/peerj.11622_

## Round 0.1 · original submission · Major Revisions

Dear Dr. Cardenas and colleagues:

Thanks for submitting your manuscript to PeerJ. I have now received three independent reviews of your work, and as you will see, the reviewers raised some minor concerns about the research. Despite this, these reviewers are very optimistic about your work and the potential impact it will have on research studying Mycetomoellerius biology snd systematics. Thus, I encourage you to revise your manuscript, accordingly, taking into account all of the concerns raised by both reviewers.

While the concerns of the reviewers are relatively minor, this is a major revision to ensure that the original reviewers have a chance to evaluate your responses to their concerns. There are many suggestions, which I am sure will greatly improve your manuscript once addressed.

Please note that reviewers 2 and 3 have included marked-up versions of your manuscript.

I look forward to seeing your revision, and thanks again for submitting your work to PeerJ.

Good luck with your revision,

-joe

·

Basic reporting

This manuscript is describing a new species of the fungus-growing ant genus Mycetomoellerius using a wide variety of techniques: phylogenetics, behaviour, chemistry and morphology. This piece should be used as an example of how modern taxonomy should be done!

Kind regards
Pepijn Kooij

Basic reporting
The manuscript is well written and follows the journals structure with only few spelling corrections are needed. Introduction and background are mostly sufficient, however, I would like to suggest the authors to introduce the junior synonym M. balboai already in the introduction. The figures are relevant and of sufficient quality, however, I did detect one issue. It seems that the figure legends for Fig 3 and Fig 4 have been mixed up. All raw data has been made available, however, for the sequence data there is only the final edited and concatenated alignment available. It is unclear to me from the methods whether there were any ambiguous regions the authors had to deal with, and if yes, how they did this.

Experimental design

Experimental design
The research is well within the scope of the journal and has well defined, relevant and meaningful research questions. I fully agree with the authors that it is important to identify and describe species. I would like to congratulate the authors on the amount of work they put into this manuscript. Their rigorous investigation with a variety of techniques and thorough analyses of the data are, in my opinion, a perfect example of what modern taxonomy should look like.

Validity of the findings

Validity of the findings
All underlying data has been provided and are robust and statistically sound. Conclusions are well stated linked to original research question & limited to supporting results.

Additional comments

General comments
L48-56 “Diverse studies … 1995 to 2019”
I suggest the authors to split this sentence in two. Finishing after “taxonomists” and continuing with “In fact, an average …”

Also, here and further throughout, I suggest the authors to consider the recently published paper on the new leaf-cutting ant genus Amoimyrmex (Cristiano et al., Austral Entomology, 2020) as one of their examples as well as in their counts of new species.

L58-62 “The lower attines … Leucoagaricus gongylophorus”
The recurring assumption among scientists in this field is indeed that the lower attines
cultivate “undomesticated” fungal cultivars, however, this wording is in my opinion incorrect. From the data so far published (e.g. the by the authors cited Mueller et al 2018), it is clear that the clades of fungi grown by lower attine ants are predominately domesticated with only a few exceptions of “free-living” taxa/specimens. Due to the low frequency of these free-living specimens, this phenomenon could easily be explained by rare occasions of fungi escaping the mutualism. Furthermore and in light of this, I would still consider these fungi as obligately mutualistic as they normally not able to live without the ants, i.e. the free-living taxa/specimens seem to be different species based on the published phylogenies.

L79-82 “It is this … (Appendix Table 8).”
This is the first table referred to in the text, I suggest the authors to number the tables, also those in the appendix, by order of appearance.

L104-105 “Based on … Mycetomoellerius mikromelanos sp. nov.”
Because this is the first appearance of the name Mycetomoellerius mikromelanos in the main text I suggest the authors to add the citation to the name.

L112-113 “Colonies of … Republic of Panama”
Please, add the total number of colonies collected/used here.

L162 “We set k = 2 … iterations.”
I am wondering whether the authors chose this number arbitrarily or performed a test to see which value of k to use. I suggest the authors to perform such a test to exclude any ambiguity.

L182-183 “We used … same colony”
I have to admit that this is not entirely clear to me. Does GLMM make sure that these samples, which are technical replicates, are seen as that? If not, these values should be averaged first before being used for statistical analysis.

L299 Section “Phylogenetic analysis”
I suggest the authors to mention here as well the total number of positions of the alignment and the number of informative characters.

L371
Chance “Lichtl” into “Licht”.

L376
Chance “underround” into “underground”.

L386-388 “A name … unavailable”
The authors mention here that a previously used name for this was incorrect and unavailable. Related to this, but not in this line, I noticed that in various of the supplementary files (especially the R scripts) the name “fovouros” was used. I suggest the authors to correct this throughout the documents.

L415-416 “Pilosity and color … ferrugineous-orange”
I was wondering if there is a colour referencing method for insects. For fungi, I know we use the following publication to identify the colour and give it a number code “Kornerup, A., Wanscher, J.H., 1978. Methuen Handbook of Colour. Eyre Methuen, London.”

L574-576 “Escovopsis, … M. mikromelanos.”
While it is commonly assumed that Escovopsis is a parasite, the published evidence in the cited references is far from strong, mostly because this was only tested on young colonies only. Among some of my colleagues we are starting to believe this is an opportunistic commensal fungus that can switch mode to parasitism (following the mutualism-commensal-parasitism continuum) when conditions are right. It is quite striking almost never adult colonies are found severely infected. I suggest the authors chance this to “an assumed micro-filamentous fungal parasite” or similar.

L834-842 “For example, … higher attines”
I have several concerns regarding the cited publication. I helped review the paper in question and feel that our concerns were not taken seriously. The genetic data in the cited paper has been manipulated (completely removing ambiguous regions) in such a way that it lost a lot of species signals in there. Ambiguous regions should have been recoded using one of various available software packages, but comments regarding that were ignored. As a result the naming of clade A (which is a single species) and clade B are possibly misinterpreted, including the placement of some of the taxa. rDNA in these taxa behaves quite strangely, especially in the polyploid Leucoagaricus gongylophorus grown by leaf-cutting ants. The possibility of L gongylophorus having adopted some higher attine fungal nuclei is not discussed or considered. In light of this, I suggest the authors to use this citation and it’s conclusions with caution.

Legends of Fig 2 and Fig 4 (currently Fig 3)
In the legend it says “Scp = Scape” but in the figures there is “Sc”.

Appendix table of literature research
Please, correct the spelling of my name in the corresponding author column: “Pepijn Kooij”

·

Basic reporting

Thank you for the opportunity to review this manuscript. This paper deals with the description of a new species of the ant genus Mycetomoellerius based on integrative taxonomy tools. The approach employed lend firm support for the species described here. The work is competent, very well written, and pleasingly illustrated. I'm glad to recommend publication under minor revision.

Experimental design

Sadly, I'm not familiar with most of the analyses and techniques employed in this manuscript, but I'm familiar with ant taxonomy and morphology. This is a great contribution to ant taxonomy, especially regarding the classification of fungus-farmings ants.

Validity of the findings

The paper is certainly original. All underlying data have been provided; they are robust, statistically sound & controlled.

Additional comments

I am happy to recommend publication, subject to the author's consideration of the minor points raised in the revised version of the manuscript.

·

Basic reporting

The English language is clear. Sufficient literature references are provided. The article structure, text, figures, tables, and all supplementary files comply with the journal standard.
I suggest just minor revisions that don't need to be re-reviewed.
- re-evaluate the use of the terms gyne and queen (text and figures)
- improve the introduction presenting the genus species number, and more details of the history of M. zeteki taxonomy and its occurence
- provide dorsal imagens of all castes
- change the epithet fovouros for mikromelanos (supplementary files)

Experimental design

The entire research project is sufficiently detailed and directed to answer the research question.

Validity of the findings

The findings have scientific validity and suitability, and contribute to the construction of knowledge about the genus Mycetomoellerius and to the use of a multiple approach for the investigation of species potentially involved in groups of cryptic species.

Additional comments

I would like to thank the authors for reviewing this impressive manuscript, which use an integrative approach to overcome the challenges that cryptic species imposes to morphological taxonomy. It should be noted that all the manuscripts of the original descriptions cited taxa are listed in the references. This is not common, but extremely important for the recognition of basic sciences such as taxonomy, especially due to the funding metrics. This recognition of taxonomy is also highlighted in the manuscript when the deposition of vouchers in well-curated and accessible natural history museum collection and the importance of species identification are mentioned.

---

## Round 0.2 · accepted · Accept

Dear Dr. Cardenas and colleagues:

Thanks for revising your manuscript based on the concerns raised by the reviewer. I now believe that your manuscript is suitable for publication. Congratulations! I look forward to seeing this work in print, and I anticipate it being an important resource for groups studying Mycetomoellerius biology and systematics. Thanks again for choosing PeerJ to publish such important work.

Best,

-joe

·

Basic reporting

no comment

Experimental design

no comment

Validity of the findings

no comment

Additional comments

I gladly recommend the paper for acceptance now. The authors have worked well on addressing the comments from myself as well as those of the other two reviewers.

I still have a personal concern with the following statement at L868-870: "For example, higher attine ants
grow two clades of fungi according to several studies (Solomon et al., 2019; Schultz et al., 2015; Jesovnik et al., 2017; Shik et al., 2021; Mueller et al., 2018)."

I disagree with the notion that higher attines grow two clades of fungi. The main single clade includes L. gongylophorus and has recently been divided into two clades without strong support (see cited references) and only a few cases of higher attines growing "lower attine fungi" have been found so far. However, this is all part of the current debate, and I accept that the authors might have different opinions regarding the interpretation of the current phylogenies. I hope I will be able to convince them otherwise in the near future with proof I am currently working on.

I believe I had mentioned this in the previous review, but I would congratulate the authors on the variety of techniques used to identify these species, because I believe that this is the way modern taxonomy should be done.

Best regards
Pepijn Kooij

·

Basic reporting

After carefully comparing the previous version with the present one and following the rebuttal letter provided step-by-step, I am convinced that authors have fully addressed all the issues pointed out in my revision. Also, I now understand that several of the problems I raised regarding the figures were artifacts of the software I was using to open them, and I apologize for that. The paper can certainly be published in this new format.

Experimental design

Ok.

Validity of the findings

Ok.

Additional comments

You have done a wonderful job joining different and informative tools to delimit this new species. Thanks for being such an inspiration for different generations of myrmecologists, old and new!